# Surveillance of Amphotericin B and Azole Resistance in *Aspergillus* Isolated from Patients in a Tertiary Teaching Hospital

**DOI:** 10.3390/jof9111070

**Published:** 2023-11-01

**Authors:** Lívia Maria Maciel da Fonseca, Vanessa Fávaro Braga, Ludmilla Tonani, Patrícia Helena Grizante Barião, Erika Nascimento, Roberto Martinez, Marcia Regina von Zeska Kress

**Affiliations:** 1Departamento de Análises Clínicas, Toxicológicas e Bromatológicas, Faculdade de Ciências Farmacêuticas de Ribeirão, Universidade de Sao Paulo, Ribeirao Preto 14040-903, Brazil; liviammf@usp.br (L.M.M.d.F.); ludmilla@fcfrp.usp.br (L.T.); phelena@usp.br (P.H.G.B.); 2Departamento de Clínica Médica, Faculdade de Medicina de Ribeirão Preto, Universidade de São Paulo, Ribeirao Preto 14040-900, Brazil; erika.nascimento@gmail.com (E.N.); rmartine@fmrp.usp.br (R.M.)

**Keywords:** *Aspergillus* spp., *Fumigati*, *Flavi*, Cyp51A, antifungal resistance, *Galleria mellonella*

## Abstract

The genus *Aspergillus* harbors human infection-causing pathogens and is involved in the complex one-health challenge of antifungal resistance. Here, a 6-year retrospective study was conducted with *Aspergillus* spp. isolated from patients with invasive, chronic, and clinically suspected aspergillosis in a tertiary teaching hospital. A total of 64 *Aspergillus* spp. clinical isolates were investigated regarding molecular identification, biofilm, virulence in *Galleria mellonella*, antifungal susceptibility, and resistance to amphotericin B and azoles. *Aspergillus* section *Fumigati* (*A. fumigatus sensu stricto*, 62.5%) and section *Flavi* (*A. flavus*, 20.3%; *A. parasiticus*, 14%; and *A. tamarii*, 3.1%) have been identified. *Aspergillus* section *Flavi* clinical isolates were more virulent than section *Fumigati* clinical isolates. Furthermore, scant evidence supports a link between biofilm formation and virulence. The susceptibility of the *Aspergillus* spp. clinical isolates to itraconazole, posaconazole, voriconazole, and amphotericin B was evaluated. Most *Aspergillus* spp. clinical isolates (67.2%) had an AMB MIC value equal to or above 2 µg/mL, warning of a higher probability of therapeutic failure in the region under study. In general, the triazoles presented MIC values above the epidemiological cutoff value. The high triazole MIC values of *A. fumigatus s.s.* clinical isolates were investigated by sequencing the promoter region and *cyp51A* locus. The Cyp51A amino acid substitutions F46Y, M172V, N248T, N248K, D255E, and E427K were globally detected in 47.5% of *A. fumigatus s.s.* clinical isolates, and most of them are associated with high triazole MICs. Even so, the findings support voriconazole or itraconazole as the first therapeutic choice for treating *Aspergillus* infections. This study emphasizes the significance of continued surveillance of *Aspergillus* spp. infections to help overcome the gap in knowledge of the global fungal burden of infections and antifungal resistance, supporting public health initiatives.

## 1. Introduction

The genus *Aspergillus* harbors numerous cosmopolitan fungal species frequently found in diverse natural habitats, especially soil and decaying organic matter. They are responsible for food spoilage, mycotoxin contamination, and various human and animal mycoses [1]. The clinical manifestations of human infections caused by species of this genus can vary from superficial mycoses to allergic conditions, asthma, and invasive aspergillosis, the latter especially in immunocompromised patients [2]. The fungal infection varies according to the pathogen versus host relationship. Further, diverse species and strains within the *Aspergillus* genus present assorted virulence patterns and resistance to commercial antifungals. Thus, the precise identification of infection-causing agents is highly relevant to selecting optimal antifungal therapy, patient management, and prevention efforts [3].

In recent decades, the use of the polyphasic approach to identify *Aspergillus* spp. has become more common. The technique combines several data and information, such as phenotypic, molecular (including genomics, proteomics, and secondary metabolites), genetic, and phylogenetic data to reach a consensus taxonomy [4]. The genus *Aspergillus* is currently divided into subgenera and sections, and the commonly isolated species from clinical samples are mostly found within the *Aspergillus* sections *Fumigati* (*A. fumigatus sensu stricto*), *Flavi* (*A. flavus*), *Terrei* (*A. terreus*), and *Nigri* (*A. niger*). Additionally, cryptic species in the section *Fumigati* (*A. udagawae*, *A. lentulus*, and *A. pseudofischeri*) are also isolated from clinical samples. They present an intrinsic resistance to antifungals, which impose an additional complication to antifungal treatment [5].

The management of aspergillosis requires early infection recognition and species identification. Depending on the severity of the infection, rapid initiation of appropriate antifungal treatment is highly recommended. Amphotericin B was previously considered first-line treatment of aspergillosis. Currently, professional societies have recommended voriconazole, isavuconazole, or even posaconazole as first-line therapy. Combination therapy with voriconazole and echinocandin is indicated for selected patient groups, although the latter should not be used as monotherapy. Itraconazole is indicated for non-invasive aspergillosis infections. The antifungal prophylaxis for patients at increased risk of invasive mold infections is the long-term administration of posaconazole or voriconazole [6].

Over the past decade, studies have reported an increase in the frequency of antifungal resistance among clinical isolates of *Aspergillus* spp. [7]. In *A. fumigatus*, triazole resistance is related to a higher MIC of at least one triazole antifungal agent based on the epidemiological cutoff values (ECV) values or clinical breakpoints (CBP) [8,9]. The emergence of azole-resistant strains has been attributed to the use of azoles in agriculture or prolonged treatments in the clinic [10,11]. The azole resistance is a complex one-health challenge that involves many stakeholders. Among the diversity of involved professionals are medical doctors and fungi researchers [7]. In 2022, the World Health Organization (WHO) ranked *A. fumigatus* among the four ‘critical threat’ pathogens due to the high perceived public health importance. This fungal prioritizing list guides research, development, and public health action [12].

Standard protocols have been established to test the antifungal susceptibility of clinically relevant microorganisms. In this context, reliable antifungal minimal inhibitory concentration (MIC) values are a priority for antifungal resistance screening and antifungal therapy guidance for patients [13]. For MIC interpretation, ECVs identifies wild-type (WT) and non-WT profile for diverse *Aspergillus* spp. against commercial antifungals [8,14]. In contrast, the CBP defines the susceptibility (susceptible, intermediate, or resistant) of *A. fumigatus* isolates against voriconazole [9]. These data are potentially helpful for antifungal treatment guidance.

The azole antifungal target is the 14-α sterol demethylase, encoded by the *cyp51A* allele, which catalyzes a crucial step in the ergosterol biosynthesis of fungi. Many azole-resistant *A. fumigatus* clinical isolates do not present mutations within the *cyp51A* allele [15]. Nonetheless, most azole-resistant strains have mutations in the *cyp51A* allele and modifications in the promoter region. The most prevalent azole-resistance-related modifications in the *cyp51A* promoter region are tandem repeats (TR) of 34 bp (TR34), 46 bp (TR46), and 53 bp (TR53), which lead to increased expression of *cyp51A.* The single point mutations in the *A. fumigatus cyp51A* allele associated with the azole-resistant strains are the Cyp51A amino acid substitutions G54, P216, M220, G138, and G448 [16]. Particular combinations of point-mutations in the *cyp51A* allele and promoter modifications are involved in multi-azole resistance, such as TR34/L98H, TR34/R65K/L98H, and TR46/Y121F/T289A. In addition, specific azole-binding amino acids that do not compromise Cyp51A activity can reduce the enzyme’s drug affinity and alter the stability and functionality of Cyp51A. This alteration makes substrate recognition more challenging, leading to varied patterns of azole resistance. These substitutions can also obstruct the azole entry channel or disturb the heme group’s position within the protein, thereby diminishing the azole’s ability to bind to the heme group effectively and permitting substrate replacement [16]. The clusters of three and five Cyp51A amino acids substitutions (‘F46Y, M172V, and D255E’; ‘F46Y, M172V, N248T, D255E, and E427K’) have been described with different profiles of azole susceptibility, albeit with notably higher azole MICs than wild-type strains of *A. fumigatus*. The strains with these amino acid substitutions also present three silent mutations that encode the Cyp51A amino acids G89, L358, and C454, which present phylogenetic importance [16,17,18]. The cluster of five Cyp51A amino acid substitutions was first described in a clinical strain isolated in 2001 [10]. Since then, 10% of described *A. fumigatus sensu stricto* isolates harbor this cluster [10,18]. 

For triazole-resistant *A. fumigatus* strains, amphotericin B (AMB) has been commonly recommended by experts as a salvage and last-line antifungal treatment of fungal infections in a hospital environment [19,20]. Little is known about the susceptibility pattern of *Aspergillus* sp. to AMB in many parts of the world [21]. To date, it is known that the resistance to AMB has been frequently observed in *A. terreus*, with low frequency in *A. flavus* [22,23] and rare frequency in *A. fumigatus* [21,24]. Although the resistance to AMB is less common than the resistance to fungistatic agents such as triazoles [19], populations of *A. fumigatus* from Korea, Canada, and Brazil have been emerging with high AMB minimal inhibitory concentration (≥2 µg/mL), becoming a significant public health concern [21,23,25,26].

Understanding the epidemiology and antifungal resistance of *Aspergillus* spp. is essential to improving prevention and patient management efforts in the medical field. In this way, we studied 64 *Aspergillus* spp. clinical isolates recovered over 6 years (2013 to 2019) from patients at the Clinical Hospital of the Medical School of Ribeirão Preto, University of São Paulo, Brazil, regarding molecular identification, antifungal susceptibility, resistance to AMB and azoles, biofilm, and virulence in *Galleria mellonella*.

## 2. Materials and Methods

### 2.1. Fungal Isolates, Conidia Production, and Ethics

Sixty-four *Aspergillus* spp. clinical isolates were included in the study. The clinical isolates were recovered over 6 years (2013 to 2019) from respiratory and other samples from patients at the Clinical Hospital of the Medical School of Ribeirão Preto, University of São Paulo, Brazil. The clinical isolates were deposited in both culture collections of Mycology Laboratory Hospital and Laboratório de Micologia Clínica (LMC)-FCFRP/USP, Brazil. The control strains were *A. flavus* ATCC204304 and *A. fumigatus* ATCC46645. The clinical isolates were also characterized and speciated by molecular methods at Laboratório de Micologia Clínica (LMC)-FCFRP/USP. The clinical isolates were freshly sub-cultured on yeast extract agar dextrose (YAG) at 37 °C for 2 days. The conidia were suspended with autoclaved double-distilled water, filtered through autoclaved Miracloth (22 to 25 μm, Merck EMD Millipore Corporation, Billerica, MA, USA), and counted under a hemocytometer. The conidia concentration was adjusted according to the needs of each experiment.

### 2.2. Molecular Identification of Aspergillus *spp.* Clinical Isolates

Genomic DNA (gDNA) was extracted according to modifications of the method described by Junghans and Metzlaff [27]. The concentration of 2 × 10^7^ *Aspergillus* spp. conidia were inoculated in 30 mL of dextrose yeast extract (YG) liquid culture medium and incubated at 37 °C at 180 rpm (Shaker Ecotron—Infors HT, Basel, Switzerland) overnight to produce mycelium, which was collected by vacuum filtration in funnel porcelain, frozen in liquid nitrogen, and crushed with the aid of a sterilized mortar and pestle. For every 40 mg of ground mycelium, 500 μL of extraction buffer (200 mM Tris—HCl pH 8.5, 250 mM sodium chloride, 25 mM EDTA, 0.5% SDS) was added. Then, an equal volume of phenol (Sigma Aldrich, St. Louis, MI, USA) and chloroform (JT Baker, Radnor, PA, USA) 1:1 (*v*/*v*) was added, and the mixture was homogenized for about 10 min in a tube shaker (Vortex Mixer—Labnet International, Edison, NJ, USA) to precipitate proteins and break down cell walls and membranes. The samples were centrifuged for 15 min at 12,000× *g* (Eppendorf, Hamburg, Germany) to sediment the precipitated proteins and cell debris. The aqueous phase was transferred to a new 1.5 mL microtube (Axygen Scientific, Glendale, AZ, USA), and the same volume of chloroform was added to remove phenol and/or chloroform residue. The samples were centrifuged again at 12,000× *g* for 5 min, and the upper aqueous phase was transferred back to a new 1.5 mL microtube where 0.54 volumes of isopropanol (JT Barker, Radnor, PA, USA) were added and gently shaken to DNA precipitation. After centrifugation at 12,000× *g* for 1 min, the pellet was washed with 70% ethanol and centrifuged again at 12,000× *g* for 1 min. The supernatant was discarded, and the ethanol residue evaporated at room temperature for 30 min. The sediment was suspended in sterilized water (Nuclease Free Water—Promega, Tokyo, Japan) and stored at 4 °C. RNA was eliminated from each sample by treatment with 300 ng/mL of RNAse (Pure Link A—Invitrogen Thermo Fisher, Waltham, MA, USA) at 37 °C for 1 h. The concentration and purity of genomic DNA were evaluated by a spectrophotometer (Implen—München, Germany) at 260 and 280 nm wavelengths. The DNA quality was observed in 1% agarose gel electrophoresis.

The clinical isolates were identified by PCR amplification and sequencing of the Internal Transcribed Spacer (ITS) region of ribosomal DNA with the primers ITS1 (5′-TCCGTAGGTGAACCTGCGG-3′) and ITS4 (5′-TCCTCCGCTTATTGATATGC-3′) [28], and calmodulin (caM) with primers cmd5 (5′-CCGAGTACAAGGAGGCCTTC-3′) and cmd6 (5′-CCGATAGAGGTCATAACGTGG-3′) [4]. The primers benA1 (5′-AATAGGTGCCGCTTTCTGG-3′) and benA2 (5’-AGTTGTCGGGACGGAAGAG-3′) [4] were used to sequence β-tubulin of a few clinical isolates. PCR was performed with the enzyme GoTaq polymerase (Promega, Tokyo, Japan), and the sequencing with ABI3100 Fluorescence Automated Frequency Detector (Applied Biosystems, Waltham, MA, USA) using the same primers. Each generated sequence was analyzed in the ChromasPro program (Technelysium Pty Ltd., South Brisbane, Australia) and compared with sequences deposited in the GenBank database using the Basic Local Alignment Search Tool (BLAST) [29]. The sequencing results were deposited in GenBank and the submission IDs are shown in the Appendix A.

### 2.3. Conidia Measurements and Biofilm Quantification

The freshly cultured conidia were observed under light microscopy (1000× magnification, N120, Coleman, Chicago, IL, USA), and the images were captured with HDCE-X5 camera/ScopImage 9.0 software. The conidia diameter measurements were performed with ImageJ version 1.53e software. The average and standard deviation of 20 measurements were performed per clinical isolate. The results were expressed in a scatter plot, and an unpaired *t*-test was performed to analyze the difference in diameter between groups. The fungal biofilm was produced in a 96-well flat-bottom plate with 0.5 × 10^6^ conidia/mL in RPMI 1640 culture medium buffered with 0.165 M 3-(N-morpholino)propanesulfonic acid (MOPS, Sigma Aldrich Chemical Corporation, Karnataka, India), pH 7.0 at 37 °C for 48 h. Subsequently, the non-adherent conidia were washed three times with sterile PBS1×. The adhered conidia were fixed with 100% methanol for 15 min. The biofilm and matrix biomass quantification were performed according to Li and collaborators [30] and Seidler and collaborators [31], respectively. Briefly, 100 μL of 0.5% Crystal Violet (Sigma Aldrich Chemical Corporation, Karnataka, India) for biomass quantification or 100 μL of 1% Safranin (Sigma, St. Louis, MI, USA) for matrix quantification were added to each well with pre-cultured biofilm and incubated for 20 min at room temperature. The dyes were removed, and the excess was washed five times with PBS 1×. The adhered cells were discolored with 200 μL of 33% acetic acid for 5 min at room temperature. One hundred microliters of the supernatant were transferred from each well to a new 96-well plate, and the absorbance was read in a plate reader (680XR—BioRad, Hercules, CA, USA). The wavelengths were 570 nm and 492 nm for biomass and matrix, respectively. The results were expressed by a correlation graph (biomass × matrix) using the Pearson method in the GraphPad Prism software (v 5.0; GraphPad Software, La Jolla, CA, USA). Data were analyzed considering a 95% confidence interval. Data were arbitrarily split into low, medium, and high biofilm producers within the absorbance range of 0.0 to 0.5, 0.501 to 1.050, and 1.051 to 2.5, respectively.

### 2.4. Galleria Mellonella Virulence Experiments

The virulence of *Aspergillus* spp. clinical isolates were characterized in the invertebrate model *G. mellonella*, according to Renwick and collaborators [32], with modifications. Groups of 10 *G. mellonella* larvae at the sixth instar of development (225 ± 25 g) were selected for each tested clinical isolate, PBS 1× control, and *naïve*. Artificial infection in *G. mellonella* larvae was performed, and the conidia concentrations were 1 × 10^6^ conidia/mL for section *Fumigati* and 1 × 10^5^ conidia/mL for section *Flavi*. The experiment was carried out for 10 days with three biological replicates, and the most significant result was expressed in a graph. The graphs and statistical analyzes by the Log-rank method (Mantel-Cox) were performed with GraphPad Prism software (v 5.0; GraphPad Software, La Jolla, CA, USA). Data were arbitrarily split into high, medium, and low virulence by considering the death of all larvae in each group at the respective intervals of zero to 4 days, 5 to 7 days, and 8 to 10 days after the fungal inoculum in each larva.

### 2.5. Antifungal Susceptibility Testing and Data Analyses

Minimal inhibitory concentration (MIC) testing was performed according to the broth microdilution method protocol M38-A2 of Clinical and Laboratory Standards Institute [33] for amphotericin B (AMB), voriconazole (VOR), posaconazole (POS), and itraconazole (ITR). Fungal isolates were grown on yeast extract agar dextrose (YAG) for conidia production at 37 °C for 48 h. A suspension of fungal conidia on distilled sterile water was filtered with a Miracloth filter and adjusted to a 5 × 10^5^ conidia/mL concentration using a hemocytometer. Final concentrations were 2.5 × 10^5^ conidia/mL, and serial antifungal dilution (0.0625 to 32 µL/mL) were diluted in RPMI 1640 culture medium buffered with 0.165 M 3-(N-morpholino)propanesulfonic acid (MOPS, Sigma Aldrich Chemical Corporation), pH 7.0 and incubated at 37 °C for 48 h. The control strain *Aspergillus flavus* (ATCC 204304) was used to validate the experiment. Data were recorded by visual observation, and the minimal inhibitory concentration (MIC) was defined as the lowest concentration of antifungal that produces inhibition. Results were expressed by reporting the MIC range, geometric mean (GM), and MIC that inhibits the growth of 50% and 90% of the clinical isolates (MIC50/MIC90). The MIC values were evaluated according to the epidemiological cutoff value (ECV) with capture ≥ 97.5% of the statistically modeled population [14]. Clinical isolates with MIC ≤ ECV were considered wild-type, while isolates with MIC > ECV were non-wild-type to antifungal, i.e., possible antifungal-resistant strain [14].

### 2.6. Amplification and Sequencing of the Promoter Region and cyp51A Locus

The promoter region and *cyp51A* gene of *A. fumigatus s.s.* were amplified by PCR with TransStart FastPfu DNA Polymerase (TransGen Biotech—Beijing, China). The primers for PCR and sequencing were PA-7 (5′-TCATATGTTGCTCAGCGG-3′) [34], P450-A2 (5′-CTGTCTCACTTGGATGTG-3′) [35] (Diaz Guerra et al., 2003), PA-5 (5′-TCTCTGCACGCAAAGAAGAAC-3′) [34], Cyp51AR2 (5′-AGTGAATAGAGGAGTGAATCC-3′), and Cyp51AR3 (5′-CCATTGCCGCAGAGATGTC-3′) [36]. The sequences were treated by the Chromas Pro^®^ program (Technelysium Pty Ltd., South Brisbane, Australia) and analyzed by the multiple alignment methods using MEGA version 11.0.8 [37] (KUMAR et al., 2018). The sequences were compared with the sequence of a wild-type strain of *A. fumigatus* (GenBank accession number AF338659.1) [34].

### 2.7. Statistical Analysis

Statistical analysis was based on the one-way analysis of variance method (one-way ANOVA), followed by Tukey’s post-test for comparative analysis of the experiments. For the virulence test, the Long-rank statistical method (Mantel-Cox) was adopted to assess the significance of the survival curves. *p* values less than 0.05 were considered significant, and the result was considered statistically different. All analyses were performed using the GraphPad Prism software (v 5.0; GraphPad Software, La Jolla, CA, USA).

## 3. Results

*Aspergillus* spp. strains were isolated by convenience sampling over 6 years (from 2013 to 2019) from tertiary hospital patients with invasive, chronic, and clinically suspected aspergillosis. Sixty-four strains were isolated from different body sites such as sputum (54.7%, *n* = 35/64), tracheal discharge (3.1%, *n* = 264), transbronchial biopsy (1.6%, *n* = 1/64), sphenoid sinus (6.2%, *n* = 4/64), ear (4.7%, *n* = 3/64), nasal mucosa (3.1%, *n* = 2/64), and other samples (4.8%, *n* = 3/64). Based on the morphological identification and DNA sequencing of the internal transcribed spacer (ITS) of ribosomal DNA region and calmodulin gene, 62.5% (*n* = 40/64) were identified in *Aspergillus* Section *Fumigati* and 37.5% (*n* = 24/64) in *Aspergillus* section *Flavi*. Four different species were determined, 62.5% (*n* = 40) of *A. fumigatus s.s.*, 20.3% (*n* = 13) of *A. flavus*, 14% (*n* = 9) of *A. parasiticus*, and 3.1% (*n* = 2) of *A. tamarii* (Appendix A).

The virulence factors (conidia size and biofilm) were investigated for all clinical isolates. The mean diameter of the conidia was 2.796 µm and 4.511 µm for *Aspergillus* Section *Fumigati* (*n* = 40) and *Flavi* (*n* = 24) clinical isolates, respectively. An unpaired *t*-test comparing the mean values of the two groups was performed using GraphPad Prism software (v 5.0; GraphPad Software, La Jolla, CA, USA), and the smallest conidial size was observed for *Aspergillus* Section *Fumigati* (*p* < 0.0001) in comparison with *Aspergillus* Section *Flavi* clinical isolates (Figure 1). 

Pearson’s correlation coefficient was used to understand the existing correlation between the data of the biofilm biomass and matrix of *Aspergillus* sections *Fumigati* and *Flavi* clinical isolates. The analysis demonstrated a significant positive correlation between biomass and matrix (*p* < 0.0001) (Figure 2A). Thus, the amount of biomass corresponds to a similar amount of matrix for each clinical isolate. Among *Aspergillus* section *Fumigati* clinical isolates, 25% (*n* = 10/40), 65% (*n* = 26/40), and 10% (*n* = 4/40) are low, medium, and high biofilm producers, respectively. Similarly, for *Aspergillus* section *Flavi* clinical isolates, 20.8% (*n* = 5/24), 66.6% (*n* = 16/24), and 12.6% (*n* = 3/24) are low, medium, and high biofilm producers, respectively (Figure 2, Appendix A). No statistical differences were found between the amount of biomass (*p* < 0.803) and matrix (*p* < 0.054) formation capacity between *Aspergillus* sections *Fumigati* and *Flavi* clinical isolates (Figure 2B,C).

The virulence of *Aspergillus* sections *Fumigati* and *Flavi* clinical isolates was determined in the alternative virulence model *Galleria mellonella*. The advantages of this model are the similarity between the defense mechanism used by larvae and the innate immune system of vertebrates, the high reproduction rate, easy maintenance in the laboratory, and the larvae survival in a wide range of temperature (18 °C to 37 °C), which facilitates the study with human pathogens [38,39]. The *G. mellonella* larvae were artificially infected with conidia of *Aspergillus* sections *Fumigati* and *Flavi* clinical isolates, and the fungal virulence profile was scored according to the larvae survival percentage. Whereas *Aspergillus* section *Fumigati* clinical isolates presented 29.6% (*n* = 8/27), 18.5% (*n* = 5/27), and 51.85% (*n* = 14/27) of high, medium, and low virulence, respectively, *Aspergillus* section *Flavi* clinical isolates presented 82.6% (*n* = 19/23) and 17.4% (*n* = 4/23) of high and low virulence in *G. mellonella* larvae, respectively (Figure 3 and Appendix A). We performed a Mantel–Cox test based on the survival experience of the subjects in the different groups being compared. The analysis showed that *Aspergillus* section *Flavi* clinical isolates were more virulent than section *Fumigati* clinical isolates (*p* < 0.0001).

The in vitro activity (MIC GM and range) of each antifungal against *Aspergillus* sections *Fumigati* and *Flavi* clinical isolates are presented in Table 1. In *Aspergillus* section *Fumigati*, the MIC90 values were 2, 1, 2, and 1 μg/mL for AMB, ITR, VOR, and POS, respectively. In *Aspergillus* section *Flavi*, the MIC90 values for AMB, ITR, VOR, and POS were 2, 1, 4, and 0.5 μg/mL (Table 1). The quality control strains were within the recommended MIC limits (Appendix A). Overall, AMB, POS, VOR, and ITR have shown MIC values above the ECV [14,32], for 4.7% (*n* = 3/64), 45.3% (*n* = 29/64), 29.7% (*n* = 19/64), and 4.7% (*n* = 3/64) of *Aspergillus* spp. clinical isolates, respectively (Appendix A). The CBP [9] indicates 47.5% (*n* = 19/40) sensitive, 15% (*n* = 6/40) intermediate, and 37.5% (*n* = 15/40) resistant *A. fumigatus s.s.* clinical isolates to voriconazole. Globally, 67.2% (*n* = 43/64) of *Aspergillus* spp. clinical isolates have shown an AMB MIC value equal to or above 2 µg/mL (Appendix A). For *Aspergillus* sections *Fumigati* and *Flavi* clinical isolates, 62.5% (*n* = 25/40) and 75% (*n* = 18/24) have shown the AMB MIC equal to or above 2 µg/mL, respectively (Figure 1, Appendix A). Among all clinical isolates, there were three *A. fumigatus s.s.* clinical isolates with MIC above the ECV to three different antifungals, ITR, POS, and VOR (LMC6011.01 and LMC6018.01); and ITR, POS, and AMB (LMC9003.01) (Appendix A). 

To elucidate the possible azole resistance mechanism involved in *A. fumigatus s.s.* clinical isolates, the promoter region and the *cyp51A* locus were sequenced and compared with the sequence of the wild-type *A. fumigatus* reference isolate (AF338659.1). In the analysis of the promoter region of *cyp51A*, 34 nucleotides were identified at position 279 bp upstream of the coding region of the gene, which coincides with the wild-type pattern of the *A. fumigatus* reference strain. Thus, tandem repeats were not identified in all *A. fumigatus s.s.* clinical isolates of this study. The Cyp51A amino acid substitutions F46Y, M172V, N248T, N288K, D255E, and E427K were globally detected in 47.5% (*n* = 19/40) of *A. fumigatus s.s.* clinical isolates. Additionally, except for two clinical isolates with the Cyp51A amino acid substitution N248K, all clinical isolates with Cyp51A amino acid substitutions presented two polymorphisms in the promoter region (−335 bp T→C and −70 bp C→T upstream *cyp51A* start codon) and three silent mutations in the *cyp51A* locus (G89, L358, and C454). In more detail, the Cyp51A amino acid substitutions M172V and N248K were found in 17.5% (*n* = 7/40) and 5% (*n* = 2/40) of *A. fumigatus s.s.* clinical isolates, respectively. The clusters of Cyp51A amino acid substitution ‘F46Y and E427K’, ‘F46Y, M172V, and E427K’, and ‘F46Y, M172V, N248T, D255E, and E427K’ were detected in 2.5% (*n* = 1/40), 2.5% (*n* = 1/40), and 20% (*n* = 8/40) of *A. fumigatus s.s.* clinical isolates, respectively (Table 2 and Appendix A). In a comparative view of the clinical isolates with Cyp51A amino acid substitution and the respective MIC values, 94.7% (*n* = 18/19) *A. fumigatus s.s.* clinical isolates presented an MIC above the ECV of one or more triazoles. For all *A. fumigatus s.s.* clinical isolates without Cyp51A amino acid substitution, 42% (*n* = 8/21) presented MIC above ECV of one or more triazoles (Table 2 and Appendix A).

## 4. Discussion

*Aspergillus* spp. are the major filamentous fungus that causes invasive infection, mainly in immunocompromised patients [40]. The identification of *Aspergillus* species was previously usually based on phenotypic features. However, molecular characterization has strongly influenced it in recent decades [4]. The precise identification of these species is necessary since the infection varies according to the pathogen versus host relationship. Regarding the pathogen, it is mainly due to the great diversity of species and strains within the genus *Aspergillus* that present an assortment of infection patterns, resistance to commercial antifungals, mainly to drugs from the polyene class (amphotericin B), azoles, and echinocandin group. Nonetheless, the clinical incidence of fungal infections, especially resistance to antifungals, is not widely known [3].

Our study identified fungal clinical isolates from Brazilian patients in the two main *Aspergillus* sections, *Fumigati* and *Flavi.* The pathogenic species here identified (*A. fumigatus s.s.*, *A. flavus*, *A. parasiticus*, and *A. tamarii*) are globally associated with superficial, subcutaneous, chronic, and systemic mycoses [26,41,42]. Fungal infection occurs through inhalation of airborne fungal propagule, although traumatic inoculation of this propagule also occurs less frequently. The conidia and spores play a crucial role in *Aspergillus* spp. infections, and their characteristics, such as pigment and size, directly impact the fungus’s pathogenicity. For instance, *Aspergillus* spp. have two pigments associated with stress resistance mechanisms, dihydroxynaphthalene melanin (DHN)-melanin and pyomelanin [43]. Regarding the size of the fungal conidia, *A. fumigatus* produces small conidia size (2–3 µm) reaching the intimate parts of the lung tissue and causing the most invasive infections. In contrast, *A. flavus* presents a large conidia size (4–6 µm), which is easily retained in the sinuses of the face, causing sinocranial infections [43]. Our study observed the larger conidia of *Aspergillus* section *Flavi* (4.5 µm) compared to the clinical isolates of *Aspergillus* section *Fumigati* (2.7 µm). Additionally, a higher virulence in *G. mellonella* larvae was observed for *Aspergillus* section *Flavi* clinical isolates. The higher virulence of *A. flavus* was previously described in *G. mellonella* and other experimental models as mice [44,45]. The experimental procedures possibly justify the high virulence of *Aspergillus* section *Flavi* clinical isolates. The artificial inoculation of *G. mellonella* larvae disrupted the initial physical barriers, easing these larger conidia’s entry. Once inside the host, the conidia of the clinical isolates of both *Aspergillus* sections *Fumigati* and *Flavi* had an equal environment to express their pathogenicity. However, despite the previous knowledge of the higher virulence [44], little is known about the specific virulence factors of *Aspergillus* section *Flavi* that cause the increased virulence compared with *Aspergillus* section *Fumigati*. The virulence factors among *Aspergillus* spp. are multifactorial. Thermotolerance, adhesins, biofilm production, toxic metabolites, and hydrolytic enzyme secretion allow for the fungal cells to overcome the host defenses and contribute to the fungal pathogenesis [46]. Thus, we investigated the biofilm formation capacity of *Aspergillus* sections *Fumigati* and *Flavi* clinical isolates. The clinical isolates were assigned to the three groups of biofilm producers (low, medium, and high). The comparative analysis with the virulence profile of each clinical isolate in *G. mellonella* larvae indicated no relationship with biofilm production. Therefore, the fungal invasion and evasion of the host immune response have a multifactorial nature, which modulates the *Aspergillus* spp. virulence and pathogenicity in high eukaryote organisms.

A crucial concern with the positive diagnosis of *Aspergillus* spp. infection (e.g., chronic invasive aspergillosis) is chronicity, difficulty in fungal eradication, and high patient mortality rate. It is partly related to emerging resistance to antifungals frequently used in clinical medicine. The resistance of several species to the available commercial antifungals shall be considered when treating the infection [2,47]. Thus, the susceptibility of *Aspergillus* sections *Fumigati* and *Flavi* clinical isolates to antifungals was studied. Amphotericin B is widely used in hospitals as a salvage and last-line drug to treat severe and urgent cases of triazole-resistant *Aspergillus* spp. infections [19,20]. In our sampling, we observed a majority of *Aspergillus* spp. clinical isolates with high MIC of amphotericin B. Although little is known about the worldwide susceptibility of *Aspergillus* spp. to AMB, up to now, the identical incidence of high AMB MIC has been described in Korea, Canada, and Brazil. Nonetheless, the reasons behind the emergence of high AMB resistance rates in these two geographic populations are still unknown [21,23,24,25,26]. A study in a southern Brazilian region showed a low incidence of high AMB MIC for *A. fumigatus* and *A. flavus* [48], which indicates a different geographical distribution pattern in Brazil. Thus, the high AMB MIC results for *Aspergillus* sections *Fumigati* and *Flavi* clinical isolates evince the inclusion of the region under study as geographically specific for this susceptibility pattern. Alarmingly, the result indicates a high probability of AMB therapeutic failure. Therefore, the physicians of the region under study should carefully consider the prescription of AMB for *Aspergillus* infection treatment.

Triazole antifungals are the first-line recommendation therapy for superficial and subcutaneous mycoses, fungal infections in hospitals, and long-term therapy [20,49]. Resistance has been reported on all continents, although the frequency of this resistance remains unknown in many regions [7,50,51,52]. Furthermore, long-term azole therapy and unrestricted use of azole compounds in the environment as fungicides in the agricultural and horticultural industry have been identified as risk factors for developing azole resistance [7,53]. The MIC of the most recommended triazoles used in the medical field, VOR, POS, and ITR, were evaluated against the *Aspergillus* sections *Fumigati* and *Flavi* clinical isolates of this study. A high presence of non-WT *Aspergillus* sections *Fumigati* and *Flavi* clinical isolates for POS was found, and it differs in European and Asian countries, which have a higher proportion of susceptible isolates [52,54,55,56]. In contrast, non-WT clinical isolates for ITR were uncommon in our region, which differs from the higher prevalence in Europe and southern Brazil [48,57]. Voriconazole is highly recommended and the first choice for treatments of aspergillosis [6,47]. Our study found a low rate of clinical isolates with a non-WT profile for VOR. Therefore, we demonstrate the utility of the existing strategy for treating *Aspergillus* infection.

To better understand the high triazole MIC among our clinical isolates, the sequence of the promoter region and *cyp51A* locus of all *A. fumigatus s.s.* clinical isolates were investigated. This gene encodes the 14-α-demethylase enzyme critical in the ergosterol biosynthesis pathway [58]. Polymorphisms in the azole-binding amino acids that do not compromise Cyp51’s activity can reduce the enzyme’s affinity for these medications. Such polymorphisms can disrupt the stability and functionality of the enzyme, making it challenging to recognize substrates and ultimately resulting in various patterns of azole resistance. Point mutations in the *cyp51* gene are extensively documented in isolates worldwide. Nonetheless, only a few studies have described these mutations within an exclusive Brazilian cohort [16,17,18,26,59]. Contrary to the hotspot substitutions G54, P216, M220, G138, and G448 already described as directly related to triazole resistance [16], our study identified the substitution of the F46Y, M172V, N248T, N248K, D255E, and E427K amino acids in the Cyp51A protein of *A. fumigatus s.s.* clinical isolates. It is important to note that most clinical isolates presented high triazole MICs, as previously described for clinical isolates with the same amino acid substitutions [16,17,18]. The same amino acid substitutions, except for N248K, were previously described in Brazilian clinical isolates [26], which indicates that they are common in our region. The N248K amino acid substitution in Cyp51A protein of *A. fumigatus s.s.* has already been related to azole resistance. However, unlike the single substitution identified in our *A. fumigatus s.s.* clinical isolates, N248K causes triazole resistance when associated with the V436A Cyp51A amino acid substitution [59].

In summary, the two main *Aspergillus* sections, *Fumigati* and *Flavi*, were isolated from Brazilian patients. A higher virulence of *Aspergillus* section *Flavi* clinical isolates was detected in the *G. mellonella* larvae infection model. Most *Aspergillus* spp. clinical isolates presented a high amphotericin B MIC, warning of a higher probability of therapeutic failure in the region under study. Furthermore, the amino acid substitutions found in Cyp51A of *A. fumigatus s.s.* clinical isolates are related to the increase in the MIC of azoles. Even so, most clinical isolates presented a WT profile for itraconazole and voriconazole, supporting them as the first therapeutic choice in controlling *Aspergillus* infections. Here, we emphasize the need for continuous surveillance of fungal infections in the hospital environment, which aids in overcoming the knowledge gap in the global fungal burden of infections and antifungal resistance, supporting public health interventions.

## Figures and Tables

**Figure 1 jof-09-01070-f001:**
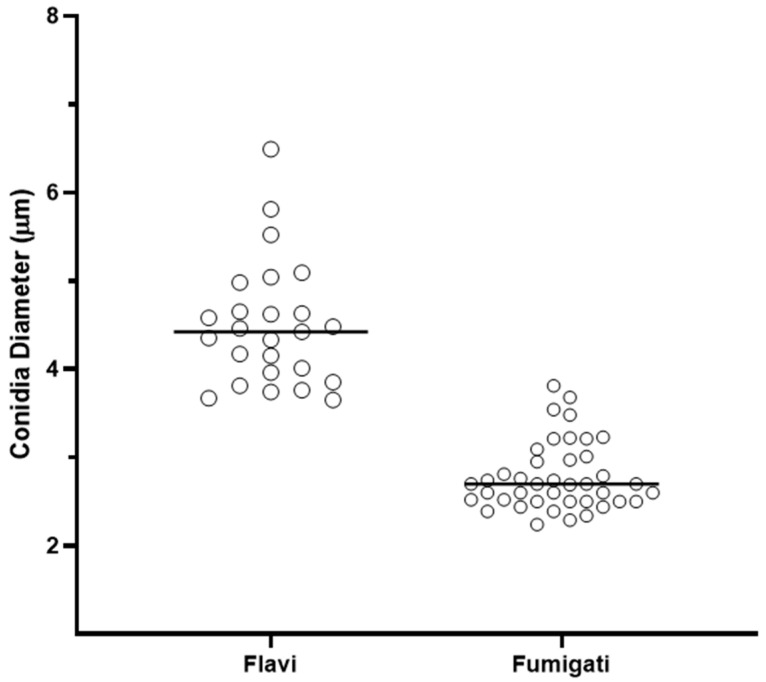
Conidia size of *Aspergillus* sections *Fumigati* and *Flavi*. Statistical analysis comparing the conidia diameter (µm) of *Aspergillus* sections *Fumigati* (*n* = 40) and *Flavi* (*n* = 24) clinical isolates (*p* < 0.0001). The bars indicate the mean and standard deviation of conidia diameter (µm). An unpaired *t*-test comparing the mean values of the two groups was performed. The *p*-value was performed with a significance level of 5% (*p* < 0.05).

**Figure 2 jof-09-01070-f002:**
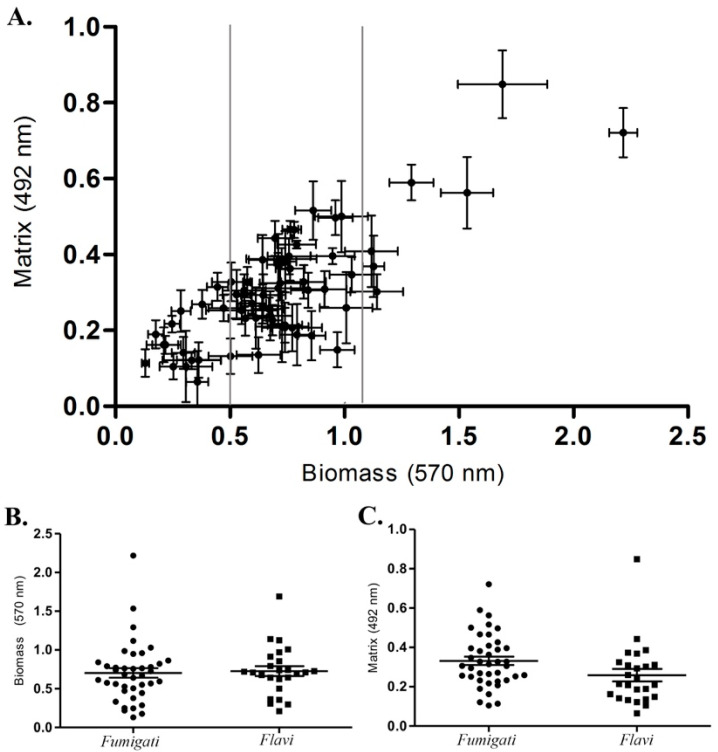
Biofilm biomass and matrix of *Aspergillus* sections *Fumigati* and *Flavi* clinical isolates. (**A**) Pearson correlation between absorbance values of biofilm biomass × matrix of 64 *Aspergillus* spp. clinical isolates and statistical analysis comparing the (**B**) biofilm biomass, *p* < 0.803 and (**C**) biofilm matrix, *p* < 0.054 of *Aspergillus* sections *Fumigati* (*n* = 40) and *Flavi* (*n* = 24) clinical isolates. The bars indicate the mean and standard deviation. The *p*-value was performed with a significance level of 5% (*p* < 0.05) by the Mann–Whitney test and is indicated in the respective graphs.

**Figure 3 jof-09-01070-f003:**
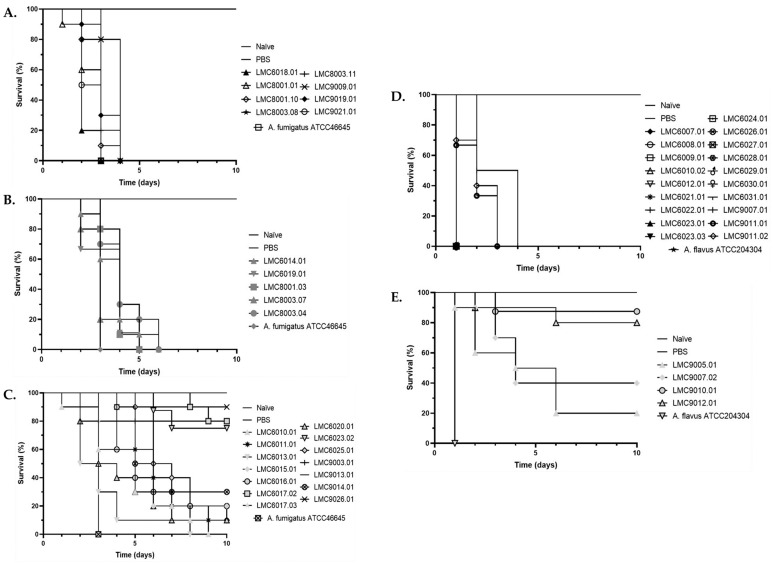
Survival of *Galleria mellonella* larvae artificially infected with conidia of *Aspergillus* sections *Fumigati* and *Flavi* clinical isolates. *Aspergillus* section *Fumigati* (**A**) high; (**B**) medium; and (**C**) low virulence, *Aspergillus* section *Flavi* (**D**) high; and (**E**) low virulence. Ten larvae were included in each group, including untouched larvae (naïve), PBS 1× control inoculation, and larvae inoculated with fungal conidia. The survival of the larvae was accompanied daily up to 10 days. PBS, phosphate-buffered saline; *p* < 0.05.

**Table 1 jof-09-01070-t001:** Minimal Inhibitory Concentration (MIC) of antifungal tested against *Aspergillus* sections *Fumigati* and *Flavi* clinical isolates.

Section	Antifungal (μg/mL)	GM	Range	MIC50	MIC90
*Fumigati* (*n* = 40)	AMB	1.625	0.5–16	2	2
	ITR	0.719	0.5–16	0.5	1
	VOR	0.949	0.25–8	0.5	2
	POS	0.42	0.03–16	0.5	1
*Flavi* (*n* = 24)	AMB	1.731	0.5–4	2	2
	ITR	0.545	0.125–1	0.5	1
	VOR	1.091	0.25–4	1	4
	POS	0.275	0.003–1	0.5	0.5

GM, geometric mean; MIC, minimal inhibitory concentration; MIC50 and MIC90, the lowest antifungal concentration able to inhibit the growth of 50% and 90% of all strains, respectively, of the tested clinical isolates.

**Table 2 jof-09-01070-t002:** Minimal Inhibitory Concentration (MIC) of antifungals and Cyp51A amino acid substitutions of *A. fumigatus s.s.* clinical isolates.

Clinical Isolate ID	Cyp51A Amino Acid Substitutions						MIC (µg/mL)			
	F46Y	M172V	N248T	N248K	D255E	E427K	ITR	POS	VOR	AMB
LMC6010.01 ^§,‡^	-	M172V	-	-	-	-	0.5	0.5 ^†^	2 ^†^	0.5
LMC6011.01 ^§,‡^	-	M172V	-	-	-	-	16 ^†^	8 ^†^	2 ^†^	1
LMC6013.01	-	-	-	-	-	-	1	0.5 ^†^	4 ^†^	1
LMC6014.01	-	-	-	-	-	-	0.5	0.5 ^†^	2 ^†^	1
LMC6015.01 ^§,‡^	F46Y	M172V	-	-	-	E427K	1	1 ^†^	4 ^†^	1
LMC6016.01	-	-	-	-	-	-	0.5	1 ^†^	2 ^†^	1
LMC6017.02 ^§,‡^	F46Y	M172V	N248T	-	D255E	E427K	1	0.5 ^†^	2 ^†^	2
LMC6017.03 ^§,‡^	F46Y	M172V	N248T	-	D255E	E427K	1	0.5 ^†^	8 ^†^	2
LMC6018.01 ^§,‡^	F46Y	M172V	N248T	-	D255E	E427K	2 ^†^	1 ^†^	2 ^†^	1
LMC6019.01	-	-	-	-	-	-	0.5	0.5 ^†^	2 ^†^	1
LMC6020.01	-	-	-	-	-	-	0.5	0.5 ^†^	1	1
LMC6023.02 ^§,‡^	-	M172V	-	-	-	-	1	0.5 ^†^	2 ^†^	0.5
LMC6025.01 ^§,‡^	F46Y	M172V	N248T	-	D255E	E427K	1	0.5 ^†^	2 ^†^	1
LMC8001.01	-	-	-	-	-	-	0.5	0.062	0.25	4 ^†^
LMC8001.03	-	-	-	-	-	-	0.5	0.125	0.5	2
LMC8001.05	-	-	-	-	-	-	0.5	0.125	1	2
LMC8001.06	-	-	-	-	-	-	0.5	0.125	0.25	2
LMC9025.01	-	-	-	-	-	-	0.5	0.125	1	2
LMC8003.01	-	-	-	-	-	-	0.5	0.25	0.5	4 ^†^
LMC8003.02 ^§,‡^	-	M172V	-	-	-	-	0.5	0.5 ^†^	0.5	2
LMC8003.05 ^§,‡^	-	M172V	-	-	-	-	0.5	0.125	0.5	2
LMC8003.06	-	-	-	-	-	-	0.5	<0.031	0.5	2
LMC8003.13 ^§,‡^	-	M172V	-	-	-	-	0.5	1 ^†^	0.5	2
LMC9026.01 ^§,‡^	-	M127V	-	-	-	-	0.5	0.5 ^†^	0.25	2
LMC9003.01 ^§,‡^	F46Y	M172V	N248T	-	D255E	E427K	8 ^†^	>16 ^†^	0.5	>16 ^†^
LMC9004.01 ^§,‡^	F46Y	M172V	N248T	-	D255E	E427K	0.5	1 ^†^	0.5	2
LMC9008.01 ^§,‡^	F46Y	M172V	N248T	-	D255E	E427K	0.5	1 ^†^	0.5	2
LMC9009.01	-	-	-	-	-	-	0.5	0.5 ^†^	0.5	2
LMC9013.01	-	-	-	-	-	-	0.5	0.25	0.5	1
LMC9014.01	-	-	-	-	-	-	0.5	0.25	0.5	1
LMC9015.01	-	-	-	N248K	-	-	1	0.5 ^†^	2 ^†^	2
LMC9016.01	-	-	-	N248K	-	-	0.5	0.5 ^†^	0.5	2
LMC9017.01	-	-	-	-	-	-	0.5	0.5 ^†^	0.5	2
LMC9018.01	-	-	-	-	-	-	1	0.25	2 ^†^	2
LMC9019.01	-	-	-	-	-	-	0.5	0.25	1	2
LMC9020.01 ^§,‡^	F46Y	M172V	N248T	-	D255E	E427K	1	0.5 ^†^	2 ^†^	2
LMC9021.01	-	-	-	-	-	-	0.5	0.25	0.5	1
LMC9022.01	-	-	-	-	-	-	0.5	0.125	1	1
LMC9023.01 ^§,‡^	F46Y	-	-	-	-	E427K	0.5	0.5 ^†^	0.5	2
LMC9024.01	-	-	-	-	-	-	1	0.5 ^†^	1	2
ATCC46645	-	-	-	-	-	-	0.5	0.25	0.5	4 ^†^

ID, identification; MIC, minimal inhibitory concentration; ^†^, MIC values > ECV (M59 protocol, CLSI and Espinel-Ingroff et al., 2018 [14]) with capture of ≥97.5% of the statistically modeled population; ^§^, silent mutations G89G, L358L, and C454C; ^‡^, nucleotides substitutions upstream *cyp51A* coding region (−335 bp T→C and −70 bp C→T); ITR, itraconazole; POS, posaconazole; VOR, voriconazole; AMB, amphotericin B. Source: Author.

## Data Availability

The data presented in this study are openly available in GenBank. For Genbank Accession number see Appendix A.

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
