# Peer review of "Surveillance of Amphotericin B and Azole Resistance in Aspergillus Isolated from Patients in a Tertiary Teaching Hospital"

_jof, 2023, doi:10.3390/jof9111070_

Round 1
Reviewer 1 Report
This is a very meaningful article. The authors studied and described the isolation, identification, biofilm and virulence of aspergillus based on clinical samples. The minimum bacteriostatic concentration and bacteriostatic effect of antifungal drugs are also important problems in clinical practice. The results of article support voriconazole or itraconazole as the first
therapeutic choice in controlling Aspergillus infections, which is a reference for the use of clinical antifungal drugs. It is hoped that the author can provide the separation and identification of aspergillus, as well as the flow diagram of antibacterial action on fungi, so that readers can better understand the overall idea of the article.
Reviewer 2 Report
Lívia et al conducted a study titled "Surveillance of Amphotericin B and Azole Resistance in Aspergillus Isolated from Patients at a Tertiary Teaching Hospital," in which they examined 64 clinically isolated Aspergillus strains. These strains were identified using ITS analysis. The study assessed factors such as biofilm biomass, virulence, and susceptibility to antifungal agents. However, it is worth noting that the data presented in the study do not lead to straightforward conclusions or offer new findings. Instead, the data appear more like records rather than substantive research findings. Furthermore, while the study provides detailed descriptions of the results, there is a lack of concise summarization of these descriptions.
Other comments as:
1. On page 134, it's important to note that identifying strains solely based on typical colony and microscopic characteristics may not be accurate. A molecular approach, as mentioned in pages 168-179, should be considered for more precise identification.
2. Regarding page 144, please make sure to pay attention to the correct notation for the concentration of 2 x 10^7 Aspergillus spp. conidia, including the correct use of superscripts.
3. In Figure 1, it would be beneficial to indicate where statistical comparisons have been performed to enhance the clarity of the results.
4. The measurement of conidial sizes is a valid question. In the Results section, the average size of Fumigati conidia is reported as 2.7 μm, but in the Discussion, it is mentioned as 3-4 μm. Clarify the reason for this discrepancy and provide an explanation for why conidial size was measured in the first place.
5. In the virulence assay, it's unclear how the authors concluded that Flavi strains were more virulent than section Fumigati. Please provide a clear rationale and evidence for this conclusion.
6. The description of azole antifungal resistance and the assay of substitutions with cyp51A amino acids is valuable, but it would be helpful to draw conclusions from these descriptions. For example, identify which substitutions are most commonly associated with azole resistance.
7. Line 330 should include italic formatting for "fumigatus," and please ensure that all species names are correctly formatted in italics throughout the manuscript.
8. Correct the statement on line 374 to indicate that the conidial size of Aspergillus fumigatus is 2-3 μm.
This paper requires significant improvement
Reviewer 3 Report
The original is well written and has an appropriate methodology and data analysis. It emphasize the need for surveillance of fungal infections in the hospital environment, filling a knowledge gap on regards fungal burden of infections and antifungal resistance.
On page 11 - some sentences dont have the term Aspergillus in italics.
End of comments.
Reviewer 4 Report
Resistance against antifungal compounds increases and be a challenge for treatment of clinical important fungi. The article describe clinical Aspergillus isolates of several species showing azole or amphotericin B resistance. Sequence data show important changes of amino acid exchanges within CYP51A encoding sterol 14-alpha demethylase. MIC values were obtained and Galleria mellonella was used as model for infection studies.
Minor points
Abstract
lines 26- 27; “The Cyp51A amino acid...” The authors should check carefully the mutations, especially N288K and correct the order of the presented mutations.
Materials and Methods
Line 187: “in RPMI 1640 pH 7.0” RPMI 1640 is a cell culture medium and not adapted for the growth of fungi. Do the authors modify the medium? Modification needs to be described. PH regulation of the RPMI 1640 medium depends on cultivation in a CO2 incubator or in addition of a buffer substance. The authors should specify the details.
2.5 Antifungal susceptibility testing and data analysis Lines 217-227
The authors should present more details of the method. Medium has a strong influence for the measurement of MIC values and therefore the used medium must be described. The source and quality of antifungal compounds is missing as well as the used concentration range for each antifungal compound. The authors should give information about timepoints to evaluate MIC values and if the values were obtained photometrically or via visual interpretation.
Results
Table 1
I have some problems to understand the values in the tables. The authors should explain the definition of range. In three lines MIC50 and MIC90 values were identical. The authors should explain this and describe the limitations of measurements.
Table 2
The presented MIC values belong to which category? MIC50 or MIC90
Line 476 I suggest to give information about sequence data acc no. within the main manuscript to allow a link between the article and the data base.
Round 2
Reviewer 2 Report
Although the draft has been significantly improved, the overall aim and quality still require further effort, such as what does "A. fumigatus s.s." mean?
The language still requires further attention, such as the first sentence of Abstract
